# Anorexia Nervosa: Reduction in Depression during Inpatient Treatment Is Closely Related to Reduction in Eating Disorder Psychopathology

**DOI:** 10.3390/jpm12050682

**Published:** 2022-04-26

**Authors:** Magnus Sjögren, Rene Klinkby Støving

**Affiliations:** 1Psychiatric Center Ballerup, 2750 Ballerup, Denmark; 2Institute for Clinical Science, Umeå University, 90185 Umeå, Sweden; 3Center for Eating Disorders, Odense University Hospital, Mental Health Services in the Region of Southern Denmark, 5000 Odense, Denmark; rene.k.stoving@gmail.com; 4Research Unit for Medical Endocrinology, Department of Clinical Research, University of Southern Denmark, 5000 Odense, Denmark

**Keywords:** anorexia nervosa, weight restoration treatment, post-meal

## Abstract

Objective: Anorexia nervosa (AN) is a severe mental disorder frequently associated with high scores of depressiveness. We examined the short-term effects of inpatient treatment on depressiveness and eating disorder (ED) psychopathology using the self-rating Major Depression Inventory (MDI) and Eating Disorder Examination questionnaire (EDEq) for patients with AN. Material: Forty-nine patients with AN, all part of the PROspective Longitudinal all-comer inclusion study on EDs (PROLED), were observed over eight weeks with baseline psychometric measures, EDE-q at baseline and endpoint, and weekly MDI self-scoring. Methods: Apart from the weekly Body Mass Index (BMI) measurements, patients were assessed at baseline using the Eating Disorder Inventory (EDI) and the Symptom Check List 92 (SCL-92). Results: Inpatient treatment reduced MDI consistently over 8 weeks (Wilks Lambda = 0.59, F = 4.1, *p* < 0.01) and this reduction in MDI was positively correlated with a reduction in EDEq (r = 0.44; *p* < 0.01) during inpatient treatment. Baseline medication did not predict changes in MDI during the inpatient treatment. BMI increased from 14.9 (week 1) to 17.2 (week 8). Conclusions: Inpatient treatment of AN is associated with a reduction in depressiveness. This improvement in depressiveness scores correlates with an improvement in ED psychopathology but not with weight gain.

## 1. Introduction

Anorexia nervosa (AN) is a severe mental disorder presenting with distorted body image, severe weight loss, and an intense fear of becoming overweight [1]. Associated symptoms are restrictive food intake and compensatory behaviors such as excessive exercise or purging [2]. This frequently leads to severe medical complications as a long-term consequence [3]. Furthermore, AN is known to be the psychiatric disorder with the highest mortality among all psychiatric disorders [4,5]. As yet, there is no approved treatment and the currently recommended therapies have been graded as weak in terms of evidence [1]. These therapies include, for example, the Maudsley model of AN treatment for adults, enhanced cognitive behavioral therapy, focal psychodynamic psychotherapy, and specialist supportive clinical management, and none have been found more superior than the other [1]. Commonly, weight restoration therapy is provided, having been shown in the short term to improve prognosis [6].

Depressiveness is very common in AN and is presented as the most common comorbidity [7]. In general, the likelihood of having a comorbid diagnosis increases with the duration of illness, a phenomenon that applies to almost all psychiatric disorders [8]. The presence of comorbidities in AN is believed to significantly influence the course and mortality rate [9]. However, depressiveness may also be due to the severely malnourished state of acute AN as proposed in several studies [10,11]. In one of these studies, patients were followed longitudinally and a relationship between depression and eating disorder (ED) psychopathology scores, both declining over time, was found. However, no relation to weight change was detected suggesting that these changes in mood and ED psychopathology were not closely related to weight gain and restoration of nutritional state [11].

Other studies have suggested that there is a relation between normalization of body weight in AN and a preferential distribution of body fat in central regions, although this was not found to influence eating disorder (ED) psychopathology or psychological distress scores [12]. This accumulation of body fat in central regions that seems to occur during weight normalization treatment in AN has been suggested to bring about considerable stress in these patients, challenging their body image and making them more prone to relapse [13]. However, data to support this hypothesis is lacking. In overweight women, a correlation between depressive mood and visceral fat tissue has been reported [14], potentially reflecting a dysregulation of the hypothalamic-pituitary-adrenal axis induced by leptin [15]. But again, no similar studies have been made and no results are available on AN. Should such a relation between depressiveness and weight gain in AN be demonstrated, it could have major implications for clinical guidelines on the treatment of AN, and consequently, such guidelines would need to implement specific support and treatment of the depression during weight restoration [16].

Since weight gain is an essential and established part of the treatment of AN, and comorbid depression has been found to negatively affect weight gain during treatment of AN [17], it is essential to increase the understanding of the relation between weight restoration treatment and depression. Thus, we investigated the change in depression scores in patients with AN undergoing weight restoration as inpatients. We also assessed the relation to ED psychopathology. Based upon clinical observations of inpatients with AN, we expected depression scores to decline upon weight gain, hypothetically due to an improved nutritional state.

## 2. Materials and Methods

### 2.1. Participants

Participants in this study were all enrolled in the PROspective Longitudinal all-comer inclusion study on EDs (PROLED), which has been described previously [18]. In total, forty-nine individuals, 46 women and 3 men who all met the ICD-10 criteria for AN at the time of hospital admission [19], and were aged between 18 and 52 years, took part in this study on weight gain. The PROLED study was approved by the local ethics board (id: H-15012537; addendum 77106), and all included individuals fulfilled the following criteria:-adult individuals (age 18–65 years);-admitted to the ED unit at the Psychiatric Center Ballerup, in Denmark;-a diagnosis of an ED.

If subjects were under forced care during the time of enrollment, they were not eligible to be included. During the first years, 96% in 2016, 74% in 2017, 62% in 2018, and 68% in 2019 who were offered to be enrolled, accepted to join the study. In the capital region of Denmark, this is the only specialized ED unit for adults with AN.

All participants went through clinical examination prior to hospitalization, and medical records were reviewed for information on psychiatric comorbidities and medication.

### 2.2. Weight Restoration Treatment

The participants were undergoing multidisciplinary inpatient therapy with support for meals and rest as previously described [18].

### 2.3. Clinical and Psychometric Measures

The PROLED protocol includes validated surveys that assess both general and specific psychopathologic aspects of AN. For this study, the Major Depression Inventory (MDI) [20], Eating Disorder Examination-Questionnaire (EDE-Q) [21], Eating Disorder Inventory (EDI) [22], and the Hopkins Symptom Checklist (SCL-92) [23] were completed by each enrolled patient. Primary and comorbid diagnoses were all validated by a second, independent physician using the ICD-10 checklist [24]. The participants were weighed once a week before breakfast. For baseline characteristics of the enrolled subjects, see Table 1.

### 2.4. Measurements of Depression

Patients were asked to do the MDI self-questionnaire to rate their level of depression every week on Thursday at 1 pm local time while spending time in a relaxing environment (Table 2).

### 2.5. Eating Disorder Psychopathology

ED psychopathology, as assessed by the EDEq, was also measured a second time, at around the same time as the 8 weeks assessment for MDI, and was done in 47 of the 49 individuals who were completing the MDI. This allowed for an evaluation of the change in ED psychopathology over the same time-period in nearly all patients.

### 2.6. Statistical Analysis

All analyses were conducted using the Statistical Package for Social Sciences (SPSS) Version 27, IBM, Armonk, NY, USA. Using the MDI total score as an outcome, the smallest difference possible in the MDI was used i.e., a minimum group difference of 1 with a standard deviation of 1.5, to provide a power greater than β = 80% (type 2-error) and a significance of α = 5% (type 1-error), for a minimum of 36 cases was included. Clinical characteristics and psychometric measures were described as means and standard deviations (Table 1 and Table 2). Correlations, using Pearson correlation, between the MDI scores and potential covariates were calculated at all time points, and if correlated, they were included in the analysis of variance (ANOVA) which was conducted to evaluate the null hypothesis that there was no change in participants’ MDI scores during the inpatient treatment.

## 3. Results

### 3.1. Participants’ Baseline Clinical Characteristics and ED Psychopathology

The descriptive data of the sample are presented in Table 1. All enrolled patients were diagnosed with typical or atypical AN (*n* = 49) and their mean age was 24.31 years (SD = 7.49), and the mean baseline BMI was 14.85 kg/m^2^ (SD = 1.87). Of the 49 patients enrolled, 46 were women (94%), and the mean duration of illness was 6.47 years (SD = 7.11). Of the 49 patients, fifteen (31%) were treated with anti-depressive medication, nine (18%) with antipsychotic medication, and eight (16%) with anxiolytic medication during the 8 weeks. Over the same time period, the mean BMI increased from 14.85 (SD 1.87) to 17.16 (SD 1.77).

A presentation of the mean scores of the EDE-Q, EDI, MDI, and AQ is shown in Table 1 and Table 2. Only the EDE-q and MDI were available for the whole sample at baseline and at 8 weeks, and for MDI also weekly.

### 3.2. Depression Outcomes

At baseline, only the BMI was correlated with the MDI scores (r = 0.42; *p* < 0.01), and no correlations were found for age, duration of illness, EDE-q score to MDI, or any other factors.

The results of the repeated measures ANOVA for the MDI scores indicated a significant time effect, Wilks Lambda = 0.59, F = 4.1, *p* < 0.01, with an observed power of 0.85. There was no effect of baseline medication e.g., antidepressants, antipsychotics, or anxiolytics usage on the results of the repeated measures ANOVA.

Follow-up comparisons indicated that the following pairwise comparisons with Bonferroni correction for multiple comparisons were significant; between weeks 1 and 4 (*p* < 0.001), week 1 and week 5 (*p* < 0.001), and week 1 and weeks 6, 7, 8 respectively (*p* < 0.01).

A repeated measures ANOVA was also done on the EDEq total scores which also indicated a significant time effect with Wilks Lambda = 0.82, F = 10.1, *p* < 0.01, observed power = 0.87. As a post-hoc analysis, individual subscores were compared between baseline and 8 weeks using repeated measures ANOVA finding a significant change in restraint, eating concern (*p* < 0.01 for both), and weight concern (*p* < 0.05) but not in shape concern (ns) indicating that the changes observed in AN during inpatient weight restoration in ED psychopathology was confined mainly to restraining and eating concerns and possibly also weight concerns but not shape concerns.

Regarding the longitudinal correlations, the change over eight weeks in the MDI depression scores was positively correlated with the change in EDEq scores (r = 0.44; *p* < 0.01) while no correlation was found to the change in BMI during the same time period. Figure 1.

## 4. Discussion

The results of this study confirm what a few other studies have found, namely that inpatient treatment is associated with a reduction in depression scores. In addition, this reduction in depression scores was correlated to a reduction in ED psychopathology, which thus far only has been described by one research team before us [11]. In our study, no clear relation to an increase in weight was found, also confirming what previous studies have described [10,11]. In contrast, some studies did find an increase in depressiveness during weight gain [25], although the study objective and design differed from our study and the number of included patients was far lower in the study by Morgan et al. than in our study. In addition, the study by Morgan et al. was not powered to measure the change in depressiveness. Thereby, the implication of the results from our study and the one by Pleple et al. [11], is that inpatient weight restoration is associated with a reduction in depressiveness and, since ED psychopathology changes along with the reduction in depressiveness, varying degrees of depression are intimately integrated with the AN syndrome. The second implication is that, since no clear relation to weigh gain was found, the depression in AN may only partly be explained by the state of malnutrition.

Previous studies have also confirmed the high frequency of depression and depressive symptoms in patients undergoing weight restoration treatment at inpatient units, as demonstrated by using different assessment instruments for depression such as Beck’s depression inventory and the MDI [17,26]. Furthermore, some studies have described the alleviation of depressiveness without using antidepressant pharmacotherapy [26].

We also found that especially restraint and eating concerns in the spectrum of ED psychopathology were alleviated by inpatient weight restoration, and potentially also weight concerns although to a lesser extent. Applying a correction for the number of statistical analyses done, the alleviation in weight concern would however no longer remain statistically significant, and thus, this change in weight concern should be interpreted with caution. Interestingly, no change in shape concern was found which also replicates what previous studies have described when applying different types of treatments to AN including weight restoration [12].

There are some differences between this and previous studies [10,11], for example, that all patients in the present study were from the same catchment area, and that this study also included weekly measurements of depression scores, while the previous studies collected data from several clinical centers, and did not monitor changes weekly. However, the overall results are similar, again underscoring the close linkage between ED psychopathology and depressiveness. Our study also provides support to the relationship between extremely underweight patients and depression.

The baseline scores in the present population confirmed that depression and anxiety are among the most common comorbidities in AN [27]. Of noticeable importance, the improvement in depression scores during the hospitalization was not amplified by the pre-established pharmacologic treatment, which further underscores that inpatient treatment in itself reduces symptoms associated with AN. However, these results may only apply to short-term inpatient treatment since prolonged hospitalization may have negative consequences and has not proved to be superior to shorter stays [28].

The current study must be seen in light of the following limitations. The aim of the study was to study psychometric changes during hospitalization. Therefore, the results can not be generalized to weight changes in an outpatient setting. Moreover, drop-out is a well-recognized limitation in clinical research on patients with AN [29]. In the present study, 49 out of 75 admitted patients were eligible for the eight weeks assessment which somewhat weakened the generalizability of the findings. Three males were included, which was within the expected number, but not sufficient to identify potential gender differences.

The current study followed 49 individuals with a diagnosis of AN. The sample size was clearly sufficient to find a difference in the main outcome, the MDI, as evidenced by the sample size calculation. This sample size is similar to what was used in the study by el Gooch [12], who also explored changes in ED psychopathology after weight gain. A similar study by Mattar et al. [10], included a larger number of patients with AN but only followed them for 2 weeks while we followed AN patients for 8 weeks during inpatient weight restoration treatment, which clearly places a higher demand on retainment in view of the well-known attrition from treatment in AN [30]. The study by Pleple et al. [11] included a larger set of patients with AN and followed them for an average of almost 17 weeks. However, this study had a large variability in the duration of treatment (hospitalization), almost as high as the mean duration of treatment, and was a multicenter study which thereby adds to the variability in treatments and settings offered for AN, which may have influenced the results. Furthermore, the study by Pleple et al., only assessed the patients during the initial and last two weeks of treatment while our study followed the patients every week. Thereby, we argue that our study should present a smaller variability in a set of key variables such as all factors influencing the treatment effect, and provide a more consistent data set to enable studying trends over time. Indeed, our study did find that key variables in the MDI and BMI changed consistently, and with our study design, we were also able to study the relation to changes in the EDEq concerning the consistent changes in MDI and BMI, which makes our study unique.

Depression was assessed by the MDI, a self-assessment questionnaire, which has been validated and proved to be a reliable and sensitive method in several studies on AN [17,18]. The test-retest reliability of the MDI has been studied in a variety of different populations; however, to our knowledge, not explicitly in weight stable patients with AN. In the above-mentioned French study [10,11], other validated and recognized self-questionnaires were used, Beck’s Depression Inventory (BDI) and The Hospital Anxiety and Depression scale. The psychometric properties of the MDI and BDI were compared head-to-head in patients with primary depression and found to display good linear congruence [31]. However, depression in AN remains complex and multidimensional, and none of the established scales of depression are primarily developed for use in this population.

In this study, we related changes in depression scores to BMI because a person’s mood is known to influence adherence to treatment and prognosis. However, there are multiple factors that may affect mood during treatment. For example, patients may need to acclimatize to the environment at the beginning of the stay which may drive a beneficial effect independent of the increase in BMI.

## 5. Conclusions

Overall, the results demonstrate the beneficial effect of inpatient treatment of up to eight weeks on depression scores and ED symptoms, whereas this effect does not appear to be directly related to the weight gain itself, giving support to short-term inpatient treatment in moderate to severe AN. However, the conclusion of this study is not to neglect weight gain as a treatment goal, and the ultimate remission of the disease includes a normal body weight. In the long run, severe malnutrition leads to irreversible disability. Moreover, to develop better treatments for AN there is a need to study short-term effects, especially relevant to inpatient treatment due to the potential health economic aspects and the potential negative effects of long-term inpatient care. Depression is one of the strongest determinants of poor quality of life in AN [32] and since a substantial proportion of patients never recover, it remains important to monitor how a given treatment affects the degree of depression. In addition, relief from depression may be a prerequisite for weight restoration, and not the not the other way around [17].

## Figures and Tables

**Figure 1 jpm-12-00682-f001:**
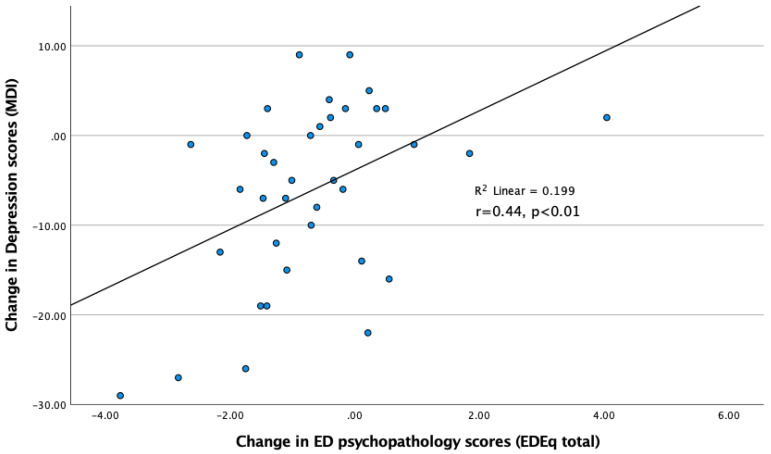
Correlation between change in Depression and change in ED psychopathology over 8 w inpatient weight-restoration treatment in AN.

**Table 1 jpm-12-00682-t001:** Participants’ (*n* = 49) baseline clinical characteristics and BMI.

Characteristics	Mean	SD
**Age (years)**	24.3	7.5
**Duration of illness (years)**	6.5	7.1
**BMI week 1**	14.8	1.7
**BMI week 2**	15.1	1.7
**BMI week 3**	15.6	1.6
**BMI week 4**	16.1	1.7
**BMI week 5**	16.4	1.8
**BMI week 6**	16.7	1.7
**BMI week 7**	16.9	1.8
**BMI week 8**	17.2	1.8
**EDEQ-Restraint**	3.4	1.6
**EDEQ-Eating concern**	3.4	1.5
**EDEQ-Shape concern**	4.8	1.3
**EDEQ-Weight concern**	4.3	1.5
**EDEQ-Global score**	4	1.3
**EDI**	174.2	57.6

**Table 2 jpm-12-00682-t002:** Participants’ (*n* = 49) Major Depression Inventory (MDI) scores over 8 weeks.

Characteristics	Mean	SD
**MDI week 1**	34	1.3
**MDI week 2**	31.8	1.4
**MDI week 3**	30.8	1.5
**MDI week 4**	28.7	1.5
**MDI week 5**	28.6	1.6
**MDI week 6**	28.6	1.7
**MDI week 7**	27.9	1.7
**MDI week 8**	27.6	1.8

## Data Availability

The datasets used in this study are available from the corresponding author.

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
