# Peer review of "Anorexia Nervosa: Reduction in Depression during Inpatient Treatment Is Closely Related to Reduction in Eating Disorder Psychopathology"

_jpm, 2022, doi:10.3390/jpm12050682_

Round 1
Reviewer 1 Report
The authors find that in a fairly small group of patients, psychological symptoms including depression and eating disorder symptoms, get better during hospitalization but there is no significant effect of weight gain. This is rather counterintuitive and might be a statistical Type 2 error related to small numbers. I did not find much discussion of this in the paper. The power calculation given did not relate to a specific variable which it should.
Author Response
We thank this peer reviewer for valuable comments and feedback. We have updated the manuscript and indicated with which outcome, i.e. MDI, that was used to calculate the sample size needed to indicate a difference. We have also added a section in the discussion on this which reads:
"
The current study followed 49 individuals with a diagnosis of AN. The sample size was clearly sufficient to find a difference in the main outcome, MDI as evidenced by the sample size calculation. This sample size is also similar to the study by el Gooch [12] which also studied changes in ED psychopathology after weight gain. A similar study, the study by Mattar et al. [10], included a larger number of patients with AN but only followed them for 2 weeks while we followed AN patients for 8 weeks during inpatient weight restoration treatment, which clearly is a higher demand on retainment in view of the well-known attrition from treatment in AN [30]. The study by Pleple et al. [11], included a larger set of patients with AN and followed them for an average of almost 17 weeks. However, this study had a large variability in duration of treatment (hospitalization), almost as high as the mean duration of treatment, and was a multicenter study which thereby adds to the variability in treatments and settings offered for AN, which may have influenced the results, and only assessed the patients during the initial and last two weeks of treatment while our study followed the patients every week. Thereby, we argue that our study should present a smaller variability in a set of key variables such as all factors influencing the treatment effect and a more consistent data set to enable studying trends over time. Indeed, our study did find that key variables in MDI and BMI changed consistently, and with our study design, we were also able to study the relation to changes in EDEq in relation to the consistent changes in MDI and BMI, which makes our study unique."
Reviewer 2 Report
The manuscript of Sjögren and Støving reported their exploration-revalidation study on the relationship between the reduction in depressiveness and ED score following AN treatment and their association with weight gain. Previous studies [10-12] have reported and discussed the apparent lack of association between AN treatment-related reduction of depression and weight gain. Here, the authors made a similar investigation using new clinical data collected on their site. Whilst such an approach may, to some degree, implicit reduced novelty. However, clinical data, especially in the field of psychiatry psychopathology, require re-validation in the population of different backgrounds. Hence, I consider this manuscript is worth publishing to serve as one more data source for the research community, after correcting some minor defects mentioned below.
L92, ->...to assess the depression level.
L117: variable -> factors
L134: remove this line and
Figure 2: Overlay the regression curve with the data points and put the regression formula used either in the figure or in the text. It appeared to me that the authors used a linear regression model in their study, it should be clearly stated in line.
L146: ...the depression may only partly be caused by the malnutrition, per se.
Author Response
We thank this peer reviewer for valuable comments and suggestions for changes. Below is a list of responses in relation to the suggestions made by this reviewer:
L92, ->...to assess the depression level.
- this has been changed.
L117: variable -> factors
- this has been changed.
L134: remove this line and
- done
Figure 2: Overlay the regression curve with the data points and put the regression formula used either in the figure or in the text. It appeared to me that the authors used a linear regression model in their study, it should be clearly stated in line.
- we have updated the graph and included the valuable suggestions for improvements made by this reviewer.
L146: ...the depression may only partly be caused by the malnutrition, per se.
- this has been changed.
- in addition, we have updated the language in the whole manuscript.
Reviewer 3 Report
Thank you very much for submitting your manuscript
Anorexia nervosa: Reduction in depression during inpatient treatment is closely related to reduction in eating disorder psychopathology
for review and consideration for publication in
Submitted to section: Mechanisms of Diseases, Biomarkers and Clinical Research in Eating Disorders, Journal of Personalized Medicine.
I sincerely appreciate the opportunity to review the manuscript.
This manuscript is a research article. I do believe that this topic is an important one and is certainly in the scope of the Journal of Personalized Medicine.
I have read it and made some minor comments. I know this means more work on your part, and this can be a difficult and frustrating task. The comments are minor, but I do feel it is still necessary to change or explain some more details regarding why you have chosen to do so. Therefore, it is not exclusively with regard to grammar.
The article focuses on connection between an eating disorder (anorexia) and depression. The focus is on short-term effects of 49 inpatient patient. The authors concluded that the baseline mediation “did not predict changes in MDI” during the inpatient treatment. The treatment associated with a reduction in depressiveness.
I enjoyed reading the article very carefully.
For me personally, the selection of the questionnaires raised further questions to which I could not find any answers in the article.
AQ: the cut-off is set at 26 points. The mean value and standard deviation add up to a total score of almost 30 points. Has it been statistically checked whether there is a change in the autistic vs. non-autistic area?
I find the AQ exciting, but why was it brought in? It is almost not mentioned in the article, or just why it is necessary. So I would argue that (unless otherwise stated) it is not necessary.
In the text, for example line 112, reference is made to MDI in Table 1. But I can't find those values anywhere. Could this possibly be added? Or why was it omitted, then please describe.
Why was the MDI used and not, for example, the Becks Depression Inventory BDI or the Hamilton Depression Scale HDRS? I would include here a brief rationale as to why the instrument used is better. It doesn't have to be statistical superiority, but time savings or something.
I also miss discussing AQ in the discussion section. It's been researched, why isn't it commented on? For example, the initial requirements that no autistic disorder is present could be confirmed.
I appreciate the opportunity to have read this manuscript and I thank you very much.
Stay healthy and all the best.
Greetings from Switzerland
Author Response
We much appreciate the comments and feedback from this peer reviewer. We have below responded to each comment from this peer reviewer:
AQ: the cut-off is set at 26 points. The mean value and standard deviation add up to a total score of almost 30 points. Has it been statistically checked whether there is a change in the autistic vs. non-autistic area?
- Se response below.
I find the AQ exciting, but why was it brought in? It is almost not mentioned in the article, or just why it is necessary. So I would argue that (unless otherwise stated) it is not necessary.
- We agree. AQ did not provide additional value in terms of the scientific question, hypothesis, design or any other aspect of this study, why we decided to explode this baseline assessment from this study.
In the text, for example line 112, reference is made to MDI in Table 1. But I can't find those values anywhere. Could this possibly be added? Or why was it omitted, then please describe.
- a reference has been added for MDI.
Why was the MDI used and not, for example, the Becks Depression Inventory BDI or the Hamilton Depression Scale HDRS? I would include here a brief rationale as to why the instrument used is better. It doesn't have to be statistical superiority, but time savings or something.
- we have added a sentence and a reference in the discussion which presents a validation study between MDI and BDI finding that they may be used interchangeably. The sentence reads as follows:
"In the above mentioned French study [10,11] other validated and recognized self-questionnaires were used, Beck’s Depression Inventory (BDI) and The Hospital Anxiety and Depression scale. The psychometric properties of the MDI and BDI was compared head-to-head in patients with primary depression and found to display good linear congruence [31]."I also miss discussing AQ in the discussion section. It's been researched, why isn't it commented on? For example, the initial requirements that no autistic disorder is present could be confirmed
- see response above.
Round 2
Reviewer 1 Report
The paper has been much improved. The authors state "
However, depression in AN remains complex and multidimensional, and none of the established scales of depression are primarily developed in this patient group." For this reason I suggest a sentence indicating the possibility of a Type II error, ie that the results do not prove that there is no relationship[ between BMI and depression scores.
Author Response
Dear Peer,
we thank you for your comment and suggestion. We have already inserted a sentence in the discussion starting on line 245 that states this. The sentence reads as follows:
"However, depression in AN remains complex and multidimensional, and none of the established scales of depression are primarily developed in this patient group."
We would not go as far as stating that since the instrument (here MDI) was not developed for Anorexia Nervosa (AN), it cannot be used to assess depression/depressiveness in this disorder. There are a whole range of psychiatric instruments that are being used in AN and other Eating disorders to assess comorbidity, with instruments that were primarily developed for another purpose than assessing comorbidity in Eating disorders, and it does not make sense to invalidate all this research.